# FunHoP analysis reveals upregulation of mitochondrial genes in prostate cancer

**Kjersti Rise[1]** *, **May-Britt Tessem[2,3]**, **Finn Drabløs[1]**, **Morten Beck Rye[1,3,4,5]** *

**1** Department of Clinical and Molecular Medicine, NTNU–Norwegian University of Science and Technology, Trondheim, Norway, **2** Department of Circulation and Medical Imaging, NTNU–Norwegian University of Science and Technology, Trondheim, Norway, **3** Clinic of Surgery, St. Olavs Hospital, Trondheim University Hospital, Trondheim, Norway, **4** Clinic of Laboratory Medicine, St. Olavs Hospital, Trondheim University Hospital, Trondheim, Norway, **5** BioCore—Bioinformatics Core Facility, NTNU–Norwegian University of Science and Technology, Trondheim, Norway

* morten.rye@ntnu.no (MBR); kjersti.rise@ntnu.no (KR)

**Data Availability Statement:** All relevant data are within the paper and its Supporting Information files.

**Funding:** This works was supported by the Liaison Committee between the Central Norway Regional

## Abstract

Mitochondrial activity in cancer cells has been central to cancer research since Otto Warburg first published his thesis on the topic in 1956. Although Warburg proposed that oxidative phosphorylation in the tricarboxylic acid (TCA) cycle was perturbed in cancer, later research has shown that oxidative phosphorylation is activated in most cancers, including prostate cancer (PCa). However, more detailed knowledge on mitochondrial metabolism and metabolic pathways in cancers is still lacking. In this study we expand our previously developed method for analyzing functional homologous proteins (FunHoP), which can provide a more detailed view of metabolic pathways. FunHoP uses results from differential expression analysis of RNA-Seq data to improve pathway analysis. By adding information on subcellular localization based on experimental data and computational predictions we can use FunHoP to differentiate between mitochondrial and non-mitochondrial processes in cancerous and normal prostate cell lines. Our results show that mitochondrial pathways are upregulated in PCa and that splitting metabolic pathways into mitochondrial and non-mitochondrial counterparts using FunHoP adds to the interpretation of the metabolic properties of PCa cells.

## Introduction

The prostate is an exceptional gland in the male body when it comes to metabolism, both in normal and cancerous cells. Most normal human cells use the tricarboxylic acid (TCA) cycle and oxidative phosphorylation to harvest energy from food. The prostate cells, however, have a different approach. When acetyl-CoA enters the TCA cycle and is added to oxaloacetate to become citrate, the citrate is secreted rather than oxidized [1]. The secreted citrate is an essential part of the prostatic fluid. Zinc ions are crucial in secretion, as these ions inhibit ACO2, the protein that converts citrate to isocitrate [2, 3]. When the prostate cells become cancerous, mitochondrial activity increases [1]. At the same time prostate cancer have also shown to (display/exert) the Warburg effect found in most tumors, where the tumor cells produce lactate

Health Authority (RHA) and the Norwegian University of Science and Technology (NTNU) to [MBR]; PhD position from Enabling Technologies, Norwegian University of Science and Technology (NTNU) to [KR], the European Research Council (ERC) under the European Union's Horizon 2020 research and innovation program (grant agreement No 758306) [MBT] and The Norwegian Cancer society [MBT]. The funders had no role in study design, data collection and analysis, decision to publish, or preparation of the manuscript.

**Competing interests:** The authors have declared that no competing interests exist.

even when oxygen is present, and limit the energy metabolism primarily to cytosolic glycolysis [4, 5]. These various types of activity highlight the importance of understanding the role of subcellular compartments when analyzing metabolic changes.

There are several reasons why cellular processes are divided between different compartments in the cell [6]. Individual subcellular compartments may define different microenvironments, favoring optimal activity for important enzymes, for example with respect to pH or ions, like zinc ions as mentioned above. Having processes in different compartments may also contribute to a more optimal distribution of metabolites between different pathways and prevent futile interactions, for example between anabolic and catabolic processes involving related substrates. And some metabolic processes may produce intermediates that are highly reactive or even toxic in relation to other pathways, and therefore needs to be kept in separate compartments. Therefore, the concept of subcellular compartments is important.

To include data on protein subcellular localization is therefore essential in order to achieve a good understanding of cellular metabolism. Several resources have measured or can predict subcellular protein localization. Experimentally localizations can be found for instance by using isotope-labeled C-atoms [7], antibodies and immunofluorescence [8], or mass spectrometry (MS), which is the method used by the SubCellBarCode (SubCell) resource [9]. SubCell uses cell fractionation combined with in-depth quantitative MS which is used as input to a bioinformatics pipeline. This gives a database of subcellular localizations. Another resource with subcellular localization data is the Human Protein Atlas (HPA), which integrates various omics technologies to create an open-source protein map [10].

The growing field of computational tools for predicting subcellular localizations also leads to new insights. The Bologna Unified Subcellular Component Annotator (BUSCA) can be used to predict the localization of proteins by using existing knowledge of amino acid patterns such as GPI anchors, signal and transit peptides, as well as transmembrane domains such as alpha-helices and beta-barrels. BUSCA uses multiple tools in its prediction–five for predictions based on the protein sequence and three for those based on the gene sequence. Especially when combined, experimental and predicted data can reveal new knowledge about the localizations of proteins. Our study combines data on localizations from two datasets of experimental data (SubCell and HPA) with predicted data (BUSCA) to find subcellular localizations for proteins from the genes under study. The localization data is used in metabolic pathway analysis, which puts the differential expression analysis from RNA-Seq into biological contexts.

Our group has previously developed a method for investigating functional homologous proteins (FunHoP), a tool for metabolic pathway analysis which improves biological interpretations by utilizing information from both pathway and gene expression data [11]. FunHoP uses biological pathways from the KEGG database [12, 13] in combination with visualizations in Cytoscape [14] using the KEGGScape app [15].

The default KEGG pathways shown both in Cytoscape and on the KEGG website display only one gene in each node in the network, even though many biological reactions can be catalyzed by alternative enzymes or homologs. In a series of steps, FunHoP extracts knowledge on these alternative homologs to expand the nodes into showing all the relevant genes. When all the genes are visible the user can apply different styles based on p-value or read counts and get a better understanding of how the genes are regulated. Alternative or homologous genes within a node can be regulated in opposite directions or found not to be significantly regulated. In this study we add a layer of subcellular localization to FunHoP. By using FunHoP to separate metabolic pathways into mitochondrial and non-mitochondrial counterparts we identify mitochondrial pathways showing that PCa cells tend to activate the TCA cycle for energy production, as well as mitochondrial-specific sub-pathways which provide the precursors to this activation.

## Materials and methods

An essential part of this study has been to add another layer of information on top of the improvements in pathway analysis that FunHoP could provide. Pathway data in XML format were downloaded from KEGG and run through FunHoP, which expanded all nodes with more than one gene, so that all genes were included for each node. This expansion made it possible to visualize all functional homologs within each pathway simultaneously. The expanded XML files were loaded into Cytoscape (version 3.4.0) via KEGGScape (version 0.7.0) and colored using transformed *p*-values from differential expression, using a red color range for downregulated genes and a green color range for upregulated genes. In this study the differential expression was calculated using RNA-Seq raw reads from two normal prostate cell lines (RWPE and PrEC) and two PCa cell lines (LNCaP and VCaP). Raw RNA-Seq SRA files were downloaded from Gene Expression Omnibus with accession GSE25183 [16]. Raw RNA-Seq reads were mapped to the hg19 transcriptome using TopHat2 [17], and featureCounts [18] was used to assign the reads to each gene. We used Voom [19] to perform differential expression analysis between prostate cancer and regular cell lines. Differentially expressed genes with a *p*-value below 0.05 were extracted, and *p*-values were log2 transformed by:

$$value = log2(p\_value) \times (-10) \times (\text{regulation})$$

where regulation was defined as 1 for upregulated genes (positive fold-change) and -1 for downregulated genes (negative fold-change) (S1 File).

The proteins created from these gene homologs have potentially different subcellular localizations, which obviously is relevant to our view of the pathways. We used three different data sources for identifying the subcellular localization of proteins from each gene (S2 File).

*SubCell*—We downloaded from SubCell ([9], https://www.subcellbarcode.org/) the subcellular localization data for the five cell lines A431, MCF7, H322, U251, and HCC827. We then made a SubCell consensus set from these data. For each gene we excluded any cell line where the localization was Unassign. If the localization was the same for all the remaining cell lines, or if one of the localizations was more frequent than the others (*i.e.*, the most frequent localization of this gene was seen more often than the second most frequent one), then this was used as the preferred localization for the gene product. Otherwise, the localization was classified as Uncertain, and this was the case for approximately 4% of the genes.

*HPA*—From Human Protein Atlas (HPA) ([10], https://www.proteinatlas.org/) we downloaded subcellular localization data based on 69 cell lines representing different tissue types. This dataset showed only minor overlap with SubCell concerning cell lines, as only MCF7 was used in both datasets. In HPA more than one localization can be assigned to each protein, and 35 different subcellular localizations are defined. Therefore, we first re-coded the annotated localizations to a simplified list of four localizations (mitochondria, cytoplasm, nucleus, and secretory), using organelle proteomes and their grouping as provided by HPA (https://www.proteinatlas.org/humanproteome/cell), but with mitochondria as a separate group. Next, we used this simplified list for each gene with more than one localization to identify a preferred localization. If the list was consistent (*i.e.*, all the simplified localizations were the same), or if one of the localizations was more frequent than the others (*i.e.*, the most frequent simplified localization of this gene was seen more often than the second most frequent one), this was used as the preferred localization for the protein. Otherwise, the localization was classified as 'uncertain', which occurred for approximately 15% of the genes.

*BUSCA*—We used BUSCA to generate predicted subcellular localization of proteins for a total of 3,341 genes with gene products belonging to the set of 85 metabolic KEGG pathways. We used Ensembl BioMart [20] to download protein sequences for this set of KEGG-relevant

genes, using HGNC [21] gene names as identifiers. Initially, we downloaded proteins for MANE Select transcripts ([22], https://www.ncbi.nlm.nih.gov/refseq/MANE/), providing one preferred transcript per gene. This gave protein sequences for 2,929 genes, or 87.7% of the relevant genes. For the remaining genes we identified the protein-coding transcript in each case that provided the longest protein sequence and used that for prediction. Finally, the complete set of sequences was used as input to the BUSCA server.

A consensus of the three localization data sets was made, using the most frequent localization in each case. For cases where each method showed a different localization (or localization was missing), the prediction from BUSCA was used, as the prediction data would be available for all genes.

For the consensus set the fraction of genes for proteins with mitochondrial localization was plotted against the number of genes (both genes in general and genes for proteins with mitochondrial localization) (S1–S3 Figs in S3 File). This was used to suggest specific pathways for further study. Four pathways were chosen as examples based on the fraction of mitochondrial proteins and previous knowledge on subcellular localization of each pathway. For each of the four chosen example pathways, the consensus was used to determine which proteins could belong to a mitochondrial version of the pathways and which proteins could not. The pathway XML files were then modified manually to create the mitochondrial and non-mitochondrial versions, which were run through the expansion part of FunHoP to get a more complete overview of the pathways. Finally, the XML files were loaded into Cytoscape for display and colored based on differential expression.

## Results

### Identification of proteins with mitochondrial localization

All the 3,341 genes from the 85 metabolic KEGG pathways were extracted to create the KEGG gene list used for the analysis. To identify proteins that localize to mitochondria, we used localization data from three sources BUSCA, HPA and Subcell. First, experimental localization data were obtained from SubCell and HPA. Since the experimental sources included localization data for only 53% (HPA) and 61% (SubCell) of the 3341 relevant KEGG pathway proteins, we also predicted subcellular localization for all the KEGG pathway genes with BUSCA, using the MANE selected transcripts if available (2,929 genes, or 87.7%), and the longest known transcript for the rest. The fraction of proteins located in mitochondria was similar in the experimental and predicted data (Table 1). The protein localization was in general quite consistent between the different methods (S1 Table in S3 File), but with some variation. The three datasets were therefore merged into a consensus set, as described under Methods.

Since a large fraction of the localizations will be based on predictions rather than experimental data the quality of these predictions will be important. To check the reliability of the predictions we compared the predicted data (BUSCA) to the experimental data from SubCell and HPA (S2 Table in S3 File), using data on whether localization was mitochondrial or not. The comparison showed an average specificity of 0.93 and sensitivity of 0.74 for the BUSCA predictions compared with SubCell and HPA, confirming that BUSCA does a prediction of mitochondrial localization that is consistent with the experimental data. We also compared the experimental data (HPA to SubCell, and vice versa) in the same way. This showed an average specificity of 0.97 and sensitivity of 0.81. Although there is somewhat better specificity and sensitivity in the experimental data, we concluded that the similarity in performance for experimental and predicted data was sufficiently robust to use the consensus localization data in the further pathway analysis.

**Table 1. Summary of localization data.**

| | Genes[a,i] | Mito[b] | %[c] | KEGG[d,i] | %[e] | KEGG & Mito[f] | %[g] |
|---|---|---|---|---|---|---|---|
| SubCellBarCode | 11613 | 737 | 6.3% | 2039 | 61.0% | 233 | 11.4% |
| Human Protein Atlas | 10837 | 819 | 7.6% | 1782 | 53.3% | 224 | 12.6% |
| Predictions (BUSCA) | | | | 3284 | 98.3% | 367 | 11.4% |
| KEGG[h] | | | | 3341 | 100.0% | | |

[a]Number of genes in dataset.

[b]Number of genes in dataset where the gene product has mitochondrial localization.

[c]Percentage with mitochondrial localization.

[d]Number of genes in dataset belonging to the set of KEGG genes.

[e]Percentage of KEGG genes.

[f]Number of genes in dataset being KEGG genes and with mitochondrial localization.

[g]Percentage with mitochondrial localization.

[h]The set of genes found in the KEGG pathways selected for analysis.

[i]Excluding genes without localization (unassigned or uncertain, see Methods).

## Mitochondrial pathways are upregulated in prostate cancer cell lines

We performed differential gene expression analysis between PCa cell lines and prostate normal cell lines and mapped the results on the metabolic genes from KEGG (S1 File). We observed a clear enrichment of upregulated genes in the mitochondria compared with other compartments. This trend was evident regardless of the resource used to determine localization (Fig 1). There also seems to be more genes outside of mitochondria that are downregulated, although this effect is less pronounced. We believe that the increase in upregulated secretory genes in the prediction data (BUSCA) compared to the experimental data may be an artefact, as most of the gene products predicted as 'extracellular' or 'membrane' by BUSCA were not in the experimental data.

Using the fraction of mitochondrial proteins in each pathway we selected four pathways as particularly interesting (Table 2). First, we selected the TCA pathway as it is known to occur inside the mitochondria. The TCA cycle has 23 out of the 29 proteins usually located inside the mitochondria and has also been shown in the literature to occur there. Second, in contrast to the first pathway we selected Glycolysis which is known to be located outside of the mitochondria. Glycolysis is generally found in the cytoplasm, and with 52 out of its 68 proteins having a non-mitochondrial localization, it was a good example of a non-mitochondrial pathway. Pyruvate metabolism and Alanine, aspartate, and glutamate metabolism were chosen as examples of pathways with a mixed localization.

All four pathway XML files were modified manually to determine how the pathways were affected by sorting the nodes according to protein localization, whether mitochondrial or non-mitochondrial. For this dataset most of the non-mitochondrial localizations were to the cytosol (59%). For TCA and Glycolysis, this modification meant removing non-mitochondrial and mitochondrial proteins, respectively. For Pyruvate metabolism and Alanine, aspartate, and glutamate metabolism, one mitochondrial and one non-mitochondrial version of each pathway was created. The following sections will look deeper into these four cases.

## The TCA cycle

The TCA cycle (Fig 2 and S4 Fig in S3 File) is a typical example of a mitochondrial pathway that is upregulated in the PCa cells. As expected, this pathway remains intact also when only mitochondrial genes are included, and nearly all the genes are upregulated in the pathway, including *ACO2*, which converts citrate to isocitrate. Thus, the combination of intact pathways

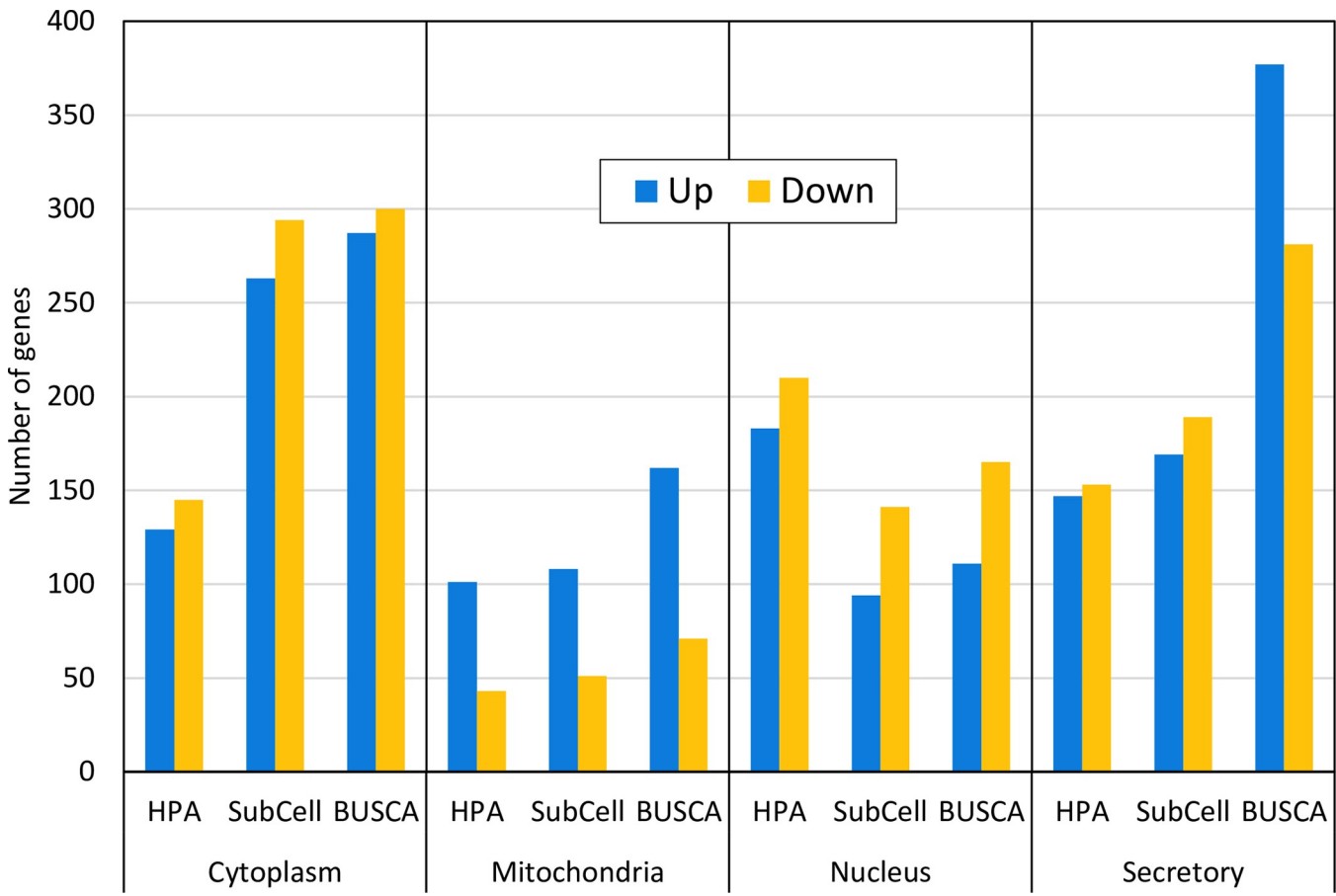

**Fig 1. Up- and downregulated genes for various subcellular localizations in the cell.** All three datasets show that the number of upregulated genes (blue bars) is higher than the number of downregulated genes (orange bars) for proteins localized to mitochondria, whereas proteins with non-mitochondrial localization are more equally distributed.

and upregulated *ACO2* confirms the current notion that the forward TCA cycle is indeed activated in PCa.

In the non-expanded TCA cycle version displayed by KEGG the mitochondrial version of the pathway is broken, since six of the proteins displayed are classified as cytoplasmic. However, five of the six proteins are really from multigene nodes, where the mitochondrial counterparts are left out. Using FunHoP to show all relevant proteins and localization layers creates awareness of all the genes included in the pathway. By adding these layers we show that the mitochondrial TCA-cycle is not broken, since the mitochondrial proteins are included in the pathway model.

**Table 2. Distribution of mitochondrial and non-mitochondrial proteins within the chosen pathways.**

| Pathway | Mitochondrial[a] | Non-mitochondrial[a] |
|---|---|---|
| TCA cycle (hsa00020) | 79% (23) | 21% (6) |
| Glycolysis (hsa00010) | 24% (16) | 76% (52) |
| Pyruvate metabolism (hsa00620) | 48% (19) | 52% (21) |
| Alanine, aspartate, and glutamate metabolism (hsa00250) | 30% (11) | 70% (26) |

[a]Shows percentage and number of genes.

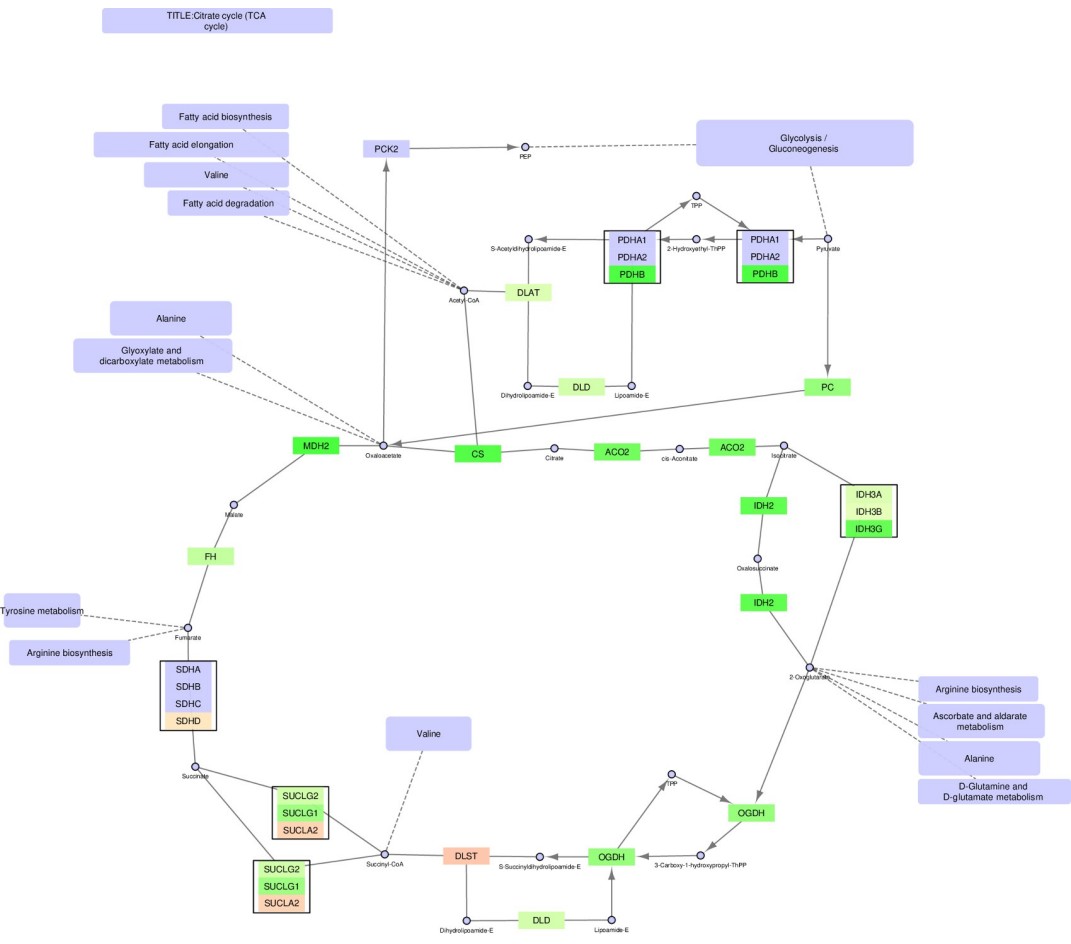

**Fig 2. The TCA cycle—mitochondrial version.**

## Glycolysis/gluconeogenesis

By contrast, Glycolysis/gluconeogenesis is an example of a pathway that is mostly non-mito-chondrial (Fig 3 and S5 Fig in S3 File).

The non-mitochondrial part of the pathway shown in Fig 3 remains connected and intact, with one exception. The original figure shows how pyruvate is converted into acetyl-CoA. As this takes place in the mitochondria, this conversion is not shown in the non-mitochondrial version, leaving pyruvate only with the conversion towards lactate. No significant upregulation of lactate dehydrogenases (*LDHA-C* and *LDHAL6A*) was found in cancer cell lines compared with normal cell lines, supporting the notion that activation of the TCA cycle is the preferred mode of energy production in the PCa cells rather than anaerobic glycolysis. The switch from *HK2* to *HK3* mediated glucose conversion in cancer cells is also interesting in this respect, since *HK2* is important in mediating the Warburg effect in prostate cancer [23].

The TCA cycle and Glycolysis/gluconeogenesis pathways remain intact in their respective compartments and serve as proof of principle that our experimental and predicted data on enzyme localization could be used to perform compartmentalized pathway analysis with the expected biological behavior. When separating pathways into mitochondrial and non-mito-chondrial counterparts, an important goal was to learn more about pathways where there is a mixture of mitochondrial and non-mitochondrial genes. We describe two pathways to

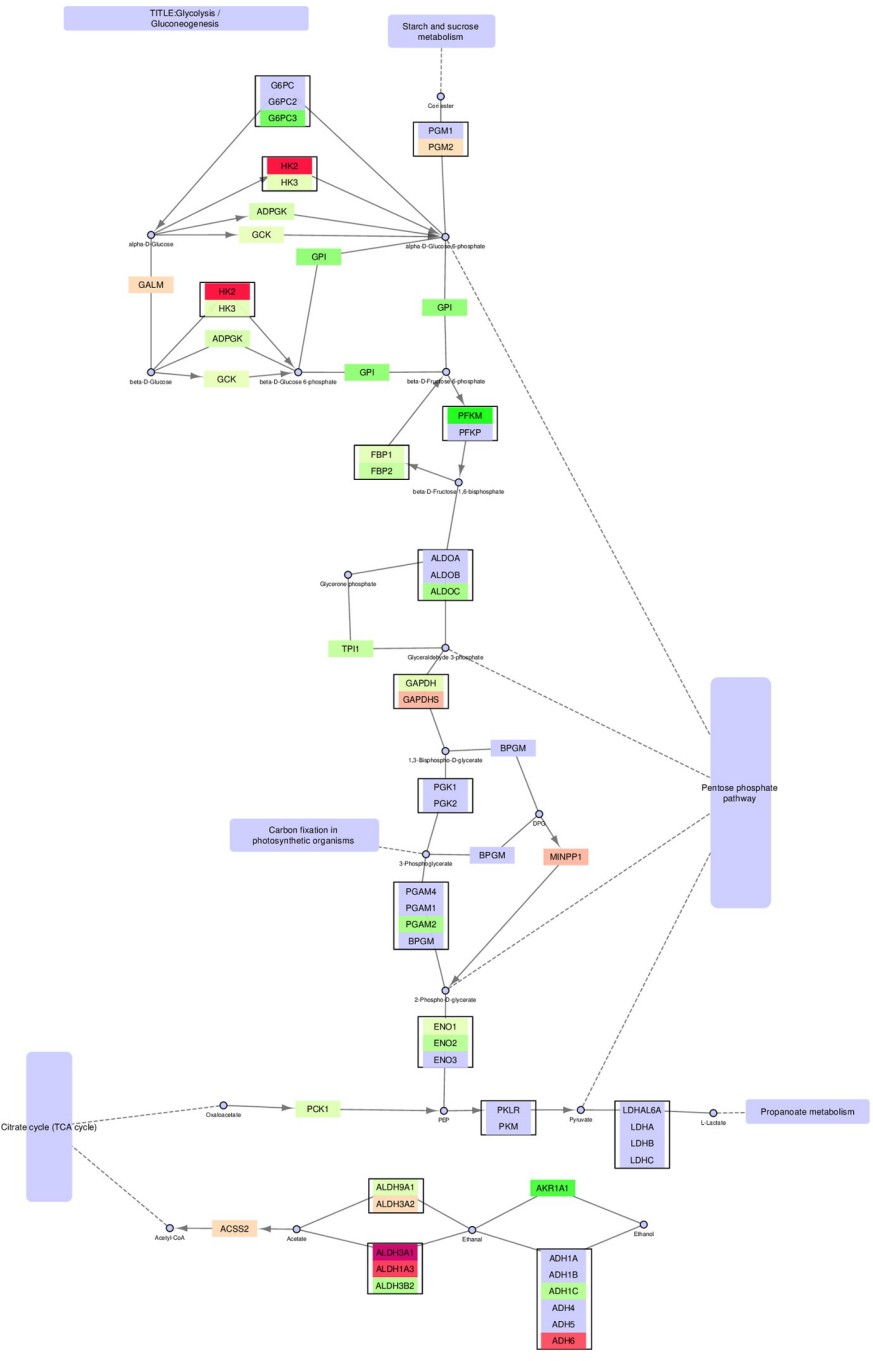

**Fig 3. Glycolysis/gluconeogenesis—non-mitochondrial version.**

exemplify this, namely Pyruvate metabolism (Figs 4–6) and Alanine, glutamine and glutamate metabolism (Figs 7–9).

## Pyruvate metabolism

The pyruvate metabolic pathway contains many genes classified as either mitochondrial or non-mitochondrial, creating substantially different pathways in these two compartments (Figs

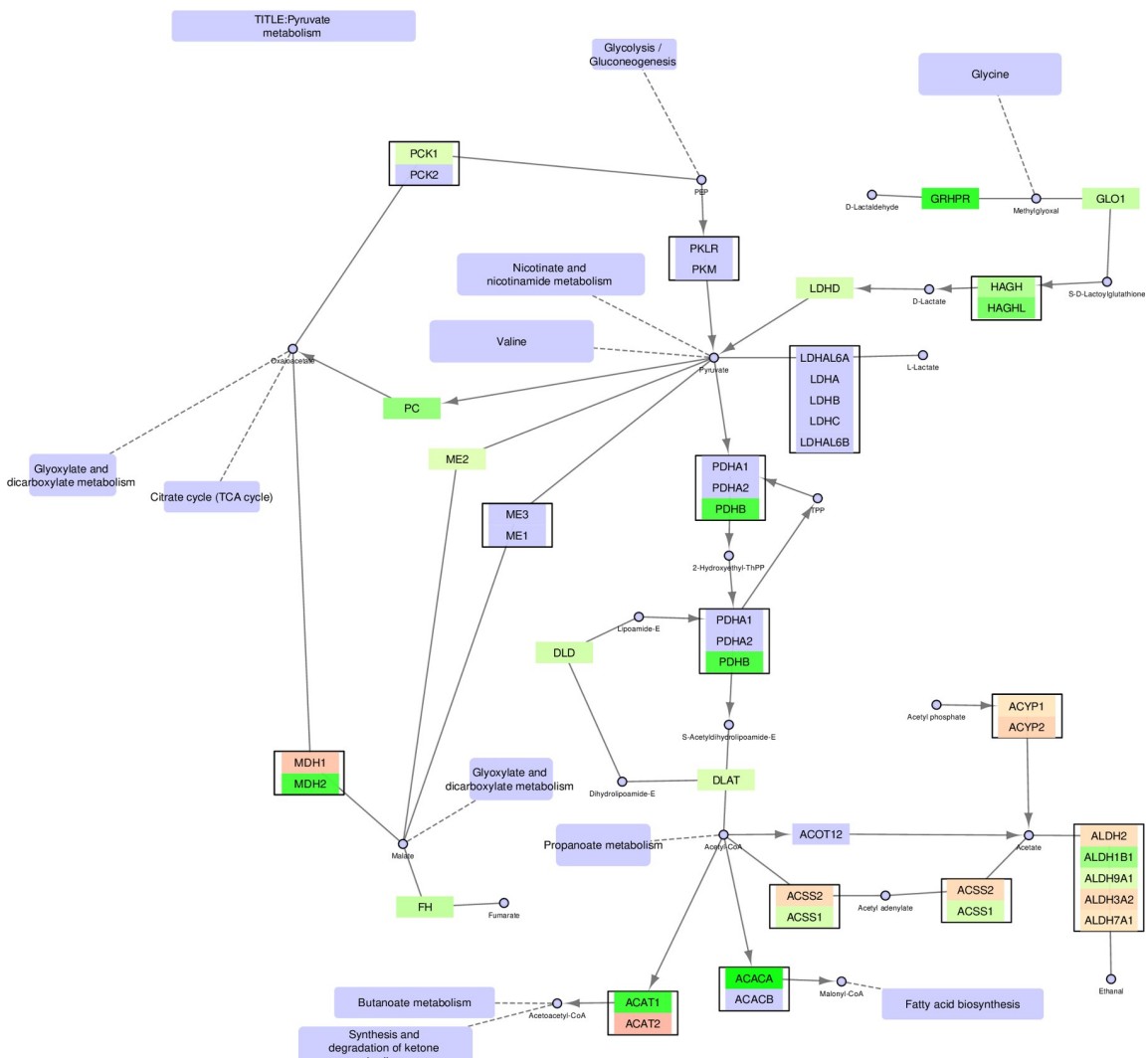

**Fig 4. Pyruvate metabolism–original version.** The FunHoP-expanded original pathway of Pyruvate metabolism is showing both up- and downregulated genes, independent of localization. The original picture changes when sorting the genes on localization into mitochondrial versus non-mitochondrial categories, and then splitting the pathway into a mitochondrial version (Fig 5) and a non-mitochondrial version (Fig 6).

4–6). The mitochondrial pathway remains continuous, with multiple branches surrounding pyruvate. In this pathway, conversion of D-lactate to pyruvate and further to oxaloacetate is increased in cancer cells by upregulation of *HAGH*, *LDHD*, and *PC*. Oxaloacetate is a precursor to and carrier of substrates in the TCA cycle, supporting the other results showing increased TCA cycle activity. The mitochondrial version also shows increased conversion of pyruvate to acetyl-CoA in cancer cells through upregulation of PDHB and DLAT.

By contrast, the non-mitochondrial version of the pyruvate pathway is broken into three separate pathways. First, there is increased conversion in cancer cells to D-lactate in the cytosol by upregulation of *HAGHL*, but no further conversions to oxaloacetate or acetyl-CoA. Instead, acetyl-CoA in the cytosol is prioritized for fatty acid synthesis by upregulation of *ACACA*. Note that the highly prioritized acetyl-CoA conversion by ACACA, leading to the activation of fatty acid synthesis is only evident when looking at the non-mitochondrial version of the

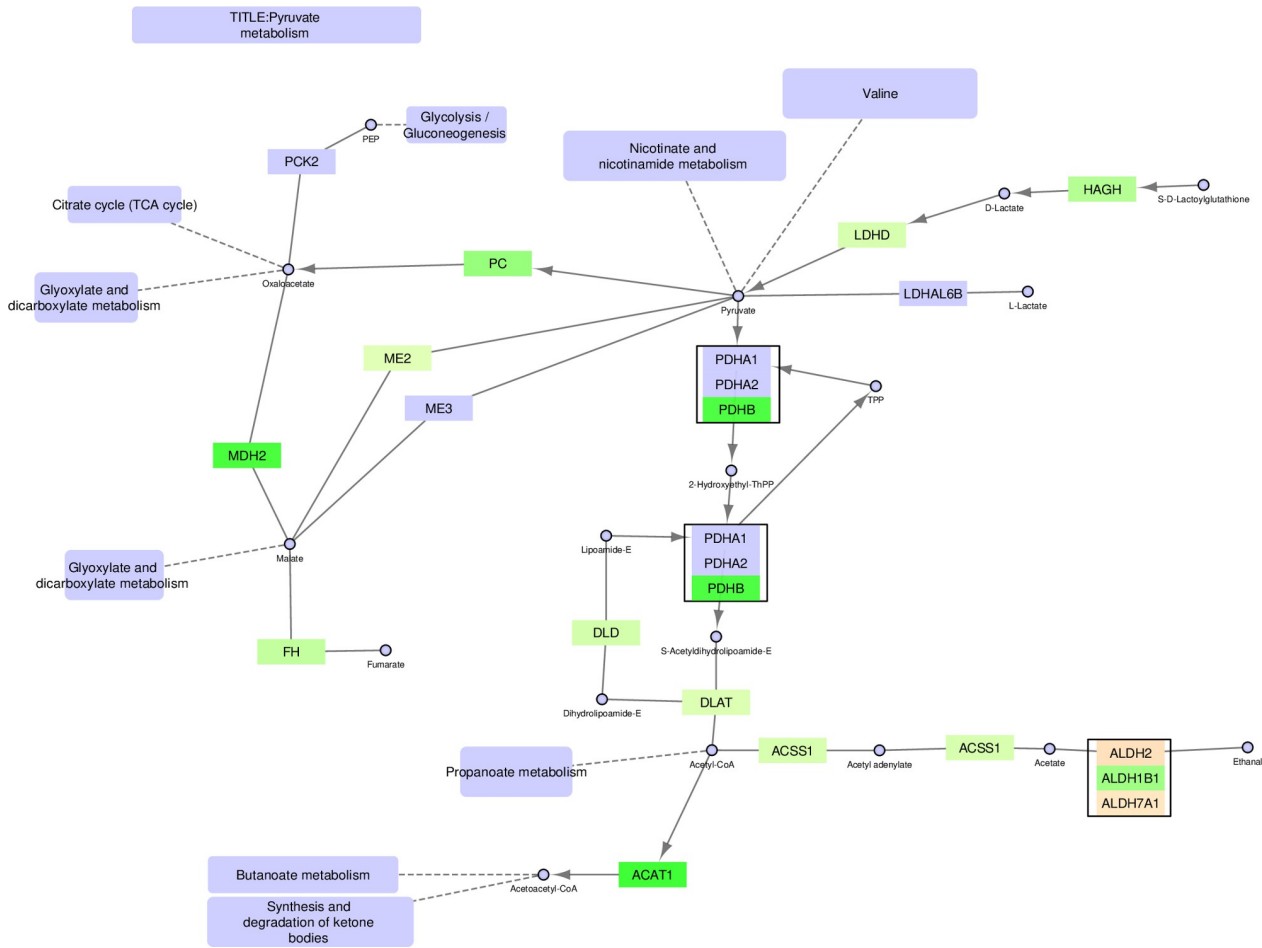

**Fig 5. Pyruvate metabolism–mitochondrial version.**

pathway, since all alternative conversion branches from acetyl-CoA are downregulated in the cancer cells. By contrast, the unmodified pathway includes a complex mixture of both up- and downregulated genes in several branches, making it more difficult to conclude on a prioritized path for acetyl-CoA. An upregulation of a smaller pathway leading towards D-lactate in the top right corner is also observed.

In the pyruvate pathway many of the two-gene nodes from the original unmodified pathway typically contain one mitochondrial and one non-mitochondrial enzyme, for example MDH1/2, ACAT1/2, and ACSS1/2. When comparing prostate cancer cell lines to normal cell lines it can often be observed that the mitochondrial genes are upregulated in cancer, while the non-mitochondrial genes are downregulated.

## Alanine, glutamine, and glutamate metabolism

Finally, the alanine/glutamine/glutamate (AGG) metabolic pathway (Figs 7–9) is examined. AGG metabolism is a complex pathway with several branches and sub-pathways surrounding the TCA cycle and providing it with metabolites affecting its activity. Again, we will focus on oxaloacetate in addition to glutamate/2-oxoglutarate.

The precursor for oxaloacetate in the AGG pathway is L-aspartate (L-Asp). Three genes can convert L-Asp to oxaloacetate, namely *GOT1*, *IL4I1* (cytosolic), and *GOT2* (mitochondrial).

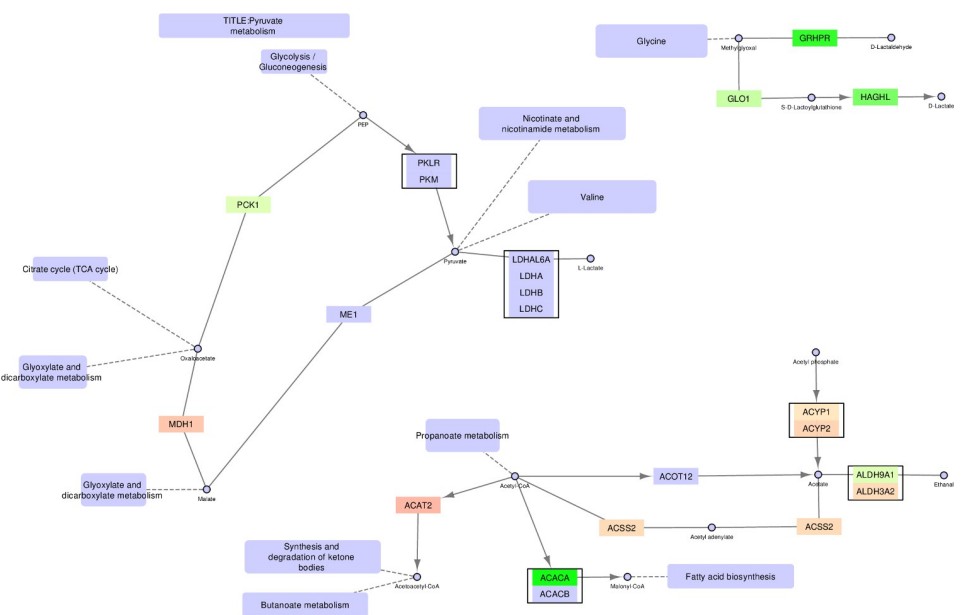

**Fig 6. Pyruvate metabolism–non-mitochondrial version.**

Of these, only *GOT1* is upregulated in PCa cells, while *GOT2* and *IL4I1* are unchanged. This indicates that the increased conversion of oxaloacetate from L-Asp in cancer occurs mainly in the cytosol and not in the mitochondria, where *GOT2* is the only enzyme acting on L-Asp. Instead, oxaloacetate can be produced in the mitochondria through the pyruvate pathway, as described previously. Alternative conversions of L-Asp are exclusive to the cytosol. Another alternative source for TCA cycle intermediates is glutamate through its conversion to 2-oxo-glutarate [24–26]. The precursor for glutamate is L-glutamine, and the path from L-glutamine via glutamate to 2-oxoglutarate is exclusive to the mitochondrial pathways through the genes *GLUD1/GLUD2* and *GLS2/GLS*. Though there is no net upregulation of these genes, a switch occurs where *GLS2* is preferred to *GLS* in the cancer cells. L-glutamine can be converted to glutamate in the cytosolic pathway (by the enzyme GLUL), but it is not converted further to 2-oxoglutarate. The latter can be produced by a separate cytosolic path, including NIT2, but it would then need to be transported to the mitochondria to be utilized in the TCA cycle.

## Discussion

### Defining the localization of gene products

When collecting the subcellular localization from the experimental data, it became clear that many genes still lack a defined localization. The two experimental datasets used in this study have a considerable overlap in 'unknowns', meaning that some data will be missing even in a combined dataset. Missing data can be complemented with tools such as BUSCA, which can be used to predict localizations based on gene or protein sequence.

A major issue when working with localization data turned out to be how to find a consensus among categories from different datasets. We decided to simplify this task in our analysis of pathways, as proteins were classified as either mitochondrial or not, and the latter were just used as a non-mitochondrial category. This was motivated by our intention to analyze the properties of mitochondrial pathways in PCa, which meant that we mainly needed a robust classification of mitochondrial versus non-mitochondrial localizations. However, the different

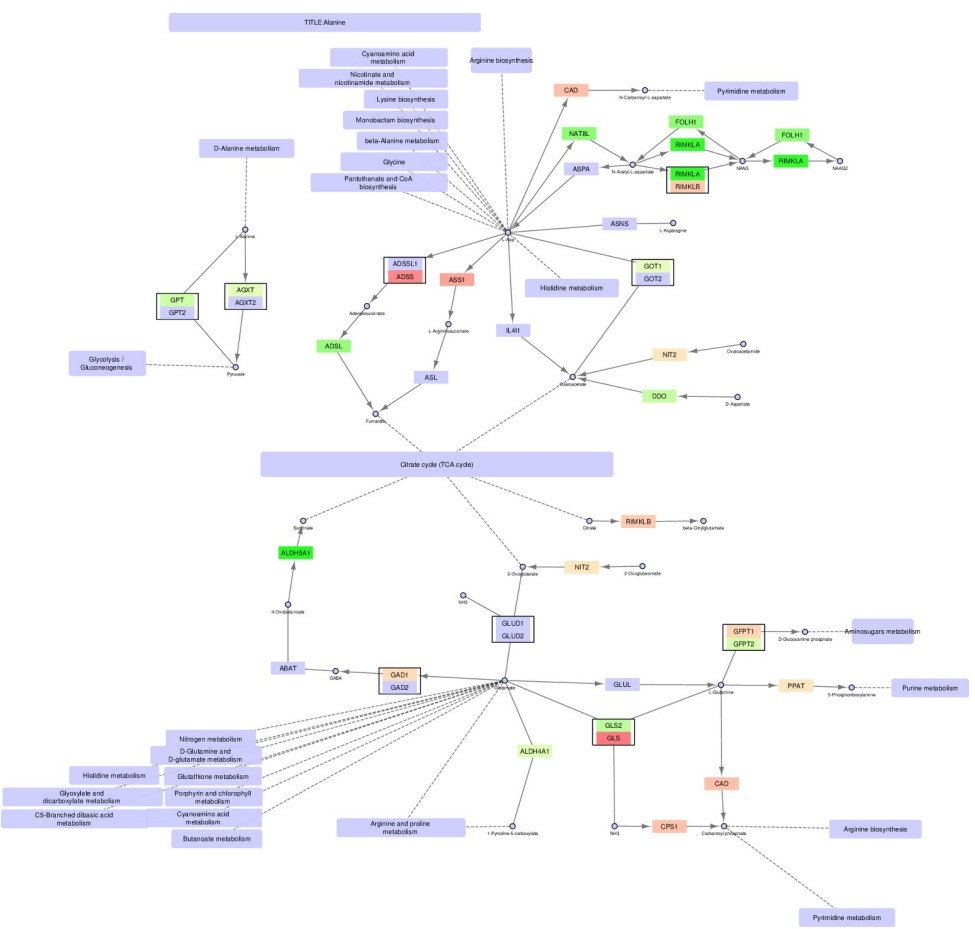

**Fig 7. Alanine, glutamine and glutamate metabolism–original version.**

resources may operate with several different localizations. The histograms in Fig 1 show how all three datasets have been reduced to a few main categories: cytoplasm, mitochondria, nuclear, and secretory, along with an 'unknown' category in the experimental data sets. For the experimental data, this reduction means that the 'secreted' group also will include extracellular–but not necessarily secreted–proteins. By contrast, the 'mitochondrial' group contains proteins found both within the mitochondria and in the mitochondrial membrane, with the equivalent solution for the nucleus. The predicted data from BUSCA does not contain an 'unknown' category but has a separate category for membranes. However, datasets from HPA and SubCellBarCode (SubCell) have both 'uncertain' (not an exact localization) and 'unknown'. A comparison of these datasets regarding 'uncertain' or 'unknown' show a considerable overlap of genes in this category. This supports our decision to use predicted localization data in addition to the experimental data.

The three datasets did not agree in all cases. This was solved by making a consensus dataset. If two out of three lists agreed, consensus went to these two, even in cases where the predictions from BUSCA differed. In cases where all three datasets differed, BUSCA was used as a casting vote. This decision was based on the number of unknown localizations from the two experimental datasets SubCell and HPA. Since BUSCA uses prediction, such predictions could be generated for all proteins with missing data in SubCell or HPA. Therefore, BUSCA was in many cases the only dataset with a suggested localization. Hence, it provided the casting vote if

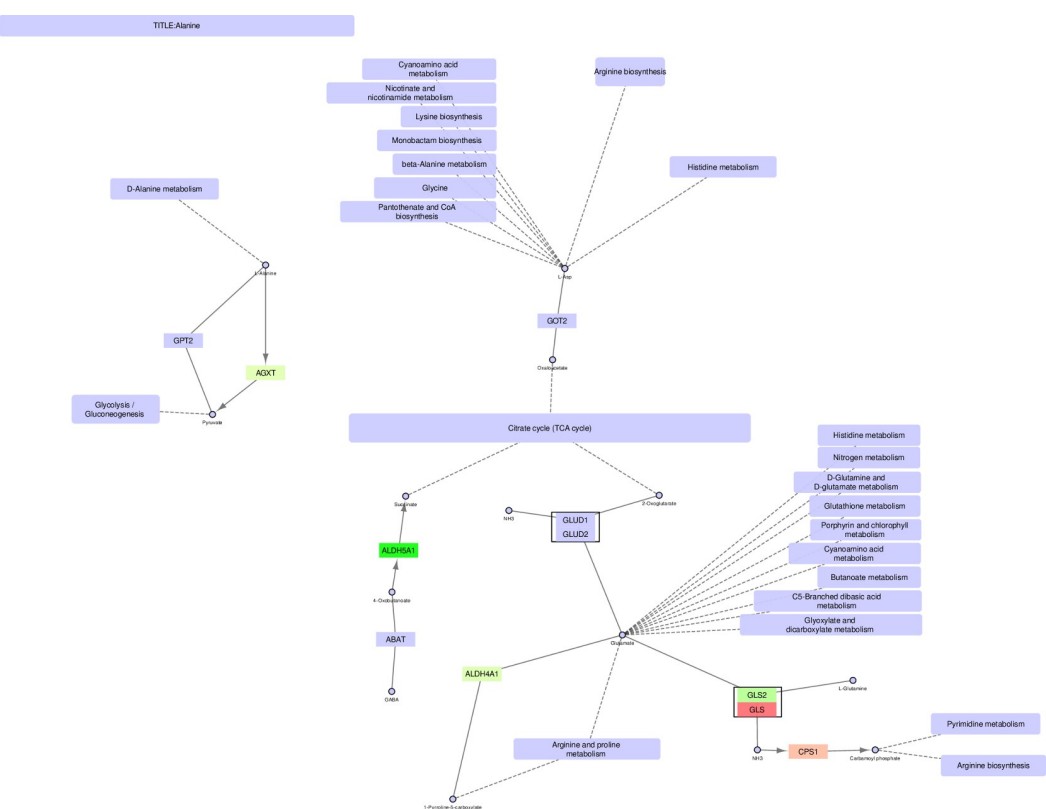

**Fig 8. Alanine, glutamine and glutamate metabolism–mitochondrial version.**

no two similar localizations could be found. However, for some proteins, such as ACACA, both experimental datasets found the localization to be cytoplasmic, while BUSCA predicted it to be mitochondrial. This shows that, although BUSCA's performance seems to be quite good (see Results), its predictions were not always consistent with other data. In cases like ACACA, with consistent experimental data, the experimental localizations would be used.

For mitochondria, most of the relevant proteins are made in the cytosol and afterwards transported into the mitochondria [27]. This means that changes in protein transport may affect the distribution of these proteins between cytosol and mitochondria. However, we do not have data that allows us to distinguish between these protein pools. Therefore, in this analysis we have assumed that the majority of the copies of each protein will be localized according to the classification described above.

## Comparison with other studies

The classification of important pathways that we see in for example Table 2 is consistent with other data. Relevant examples are the TCA cycle, which is known to be mainly mitochondrial, but linked to non-mitochondrial processes through substrates and products, whereas glycolysis is predominately a cytosolic process, but with some initiating steps taking place inside the mitochondria [28].

The general upregulation of mitochondrial activity in prostate cancer documented in this study is also consistent with previous studies. Although the original interpretation of the Warburg effect implied that oxidative metabolism (i.e., respiration) was damaged in cancer cells, several studies have shown that respiration and other mitochondrial activities are required for

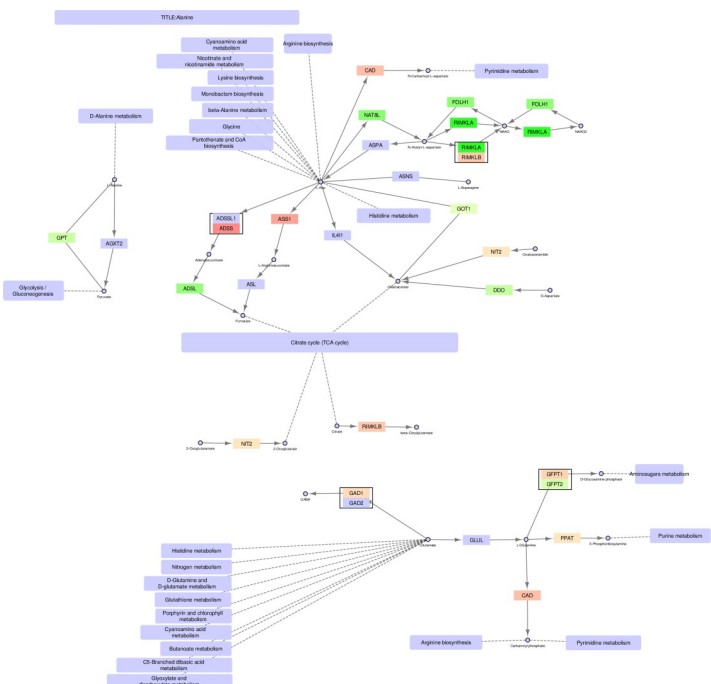

**Fig 9. Alanine, glutamine and glutamate metabolism–non-mitochondrial version.**

tumor growth (see for example [29]). Studies focusing on mitochondrial processes show in general an increase in mitochondrial activities [30–32], are relevant examples. Therefore, mitochondrial processes may also be an interesting target for treatment [33, 34].

## Interpretations of split networks using FunHoP

In this paper we illustrate how FunHoP in combination with data on subcellular localization can be used to generate more biological insights. When all multigene nodes are expanded with FunHoP, even its intermediate steps provide more information. In this study, we chose not to do a full FunHoP analysis, excluding the steps with expanded nodes and read counts, as well as the final collapsed nodes with a differential expression calculated at node level. However, using only the first and second steps of FunHoP also shows how moving from the original to the expanded view can provide the user with more biological information. For cases such as those shown in Figs 2 and 3, with paths that are expected to occur mainly in a specific localization, FunHoP can provide comprehensive information by showing all genes within each node. However, in Figs 4–9, many of the nodes with two genes had one gene found within the mitochondria and one outside (mainly cytosol), but default visualization was inconsistent in terms of which gene was displayed. If the user does not see all the genes, making a list based on only one gene in each node means dividing the paths into a mitochondrial and a non-mitochondrial pathway, which may lead to broken paths. For instance, in the TCA cycle pathway, which is usually a mitochondrial pathway, four out of five two-gene nodes with one non-mitochondrial and one mitochondrial gene showed the non-mitochondrial gene first. Without FunHoP to expand the path to show the hidden mitochondrial genes, a similar localization analysis would create an incomplete pathway with four "holes" in the path.

A notable aspect of Fig 3 is that even though most of the path is intact, starting with glucose, it stops at pyruvate, although the link to the TCA cycle is still present. This link is missing from

the non-mitochondrial version, while the original provides the conversion from pyruvate to acetyl-CoA. The removal of this conversion is a good illustration of potential improvements in pathway analysis that can be achieved by adding localizations to the analysis. Even such minor improvements can make the visualizations of pathways more biologically correct.

In our analysis of the cytosolic path including NIT2 in the AGG pathway we suggest that the metabolite 2-oxoglutarate needs to be transported to the mitochondria to be further utilized in the TCA cycle. Moreover, in terms of the split networks, some connections seem impossible unless a transport mechanism across the cellular compartments (for example, from the cytosol to mitochondria) exists. This raises an important issue regarding the role of metabolic transport mechanisms in metabolic pathway analysis. In this study we have assumed that the metabolic enzymes belong to either mitochondrial or cytosolic compartments, and thus do not move between these compartments. However, it is known that the metabolites themselves can move between compartments, with the help of transport proteins [6]. For mitochondria the solute carrier family 25 (SLC25) is important, and with 53 members it is the largest family of solute transporters in humans [35]. Other transporters or groups of transporters are also involved. Some groups of metabolites may require other transport systems, like for example lipid transport via membrane contact sites. But the main point is that knowledge about localization to subcellular compartments and transport between such compartments is important. Such transport will greatly affect the balance and availability of metabolites in the different compartments.

A necessary expansion to pathway analysis would include the integration of transport paths (with associated genes) into the networks where such paths are necessary for a reaction to occur. This information is currently not included in most pathway analyses, though its biological impact will be considerable in many cases.

## Cell lines vs. in vivo tissue data

Finally, the analysis in this study is performed on cell lines rather than on tissue samples. Cell lines used for prostate cancer research are often from metastatic cancer [36], where metabolism may be different from *in vivo*. Although cell lines in general often retain similarity to their primary tissues [37], it has also been shown that many cell lines exhibit gene expression and regulatory changes that clearly distinguish them from their origin [38], sometimes leading to only minimal similarity in biological processes [39]. Thus, the observed trends must be validated in tissue samples, for example by using publicly available datasets. However, the interpretation of metabolic pathways in tissue data can also be challenging due to heterogeneous mixtures of tissue types and other confounding factors [40]. Nevertheless, cell cultures are homogenous and work well as a model system to demonstrate proof of principle, which was the primary goal of this study.

## Conclusion

In this study, we have shown how using a combination of experimental and computational data can create a reliable consensus on the localization of gene products related to cellular metabolism. We have used this data in combination with differential expression from cancerous and normal cell lines and found that mitochondrial genes are generally upregulated in PCa cell lines. We have also shown that our program FunHoP can be used to investigate expanded networks where all relevant genes within a pathway node are shown, and that dividing such pathways into sub-pathways based on subcellular localization can provide novel biological insights.

## Supporting information

**S1 File. Differential expression analysis between LNCaP+VCaP (prostate cancer cell-lines) and RWPE+PrEC (normal prostate cell-lines).**
(XLSX)

**S2 File. Individual localization assignments from HPA, SubCell and BUSCA.**
(XLSX)

**S3 File.**
(PDF)

## Acknowledgments

We would like to thank the BUSCA team, especially Castrense Savojardo and Pier Luigi Martelli, for their help with the prediction of subcellular localization.

## Author Contributions

**Conceptualization:** Kjersti Rise, Finn Drabløs, Morten Beck Rye.

**Data curation:** Kjersti Rise.

**Formal analysis:** Kjersti Rise, Finn Drabløs.

**Funding acquisition:** May-Britt Tessem, Finn Drabløs, Morten Beck Rye.

**Software:** Kjersti Rise.

**Supervision:** Finn Drabløs, Morten Beck Rye.

**Validation:** Kjersti Rise, Finn Drabløs.

**Visualization:** Kjersti Rise.

**Writing – original draft:** Kjersti Rise.

**Writing – review & editing:** Kjersti Rise, May-Britt Tessem, Finn Drabløs, Morten Beck Rye.

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
