## [Decision Letter · Decision Letter 0]

23 May 2022

PONE-D-22-09961FunHoP analysis reveals upregulation of mitochondrial genes in prostate cancerPLOS ONE

Dear Dr. Rye,

Thank you for submitting your manuscript to PLOS ONE. After careful consideration, we feel that it has merit but does not fully meet PLOS ONE’s publication criteria as it currently stands. Therefore, we invite you to submit a revised version of the manuscript that addresses the points raised during the review process.

We look forward to receiving your revised manuscript.

Kind regards,

Thomas Patrick Burris, Ph.D.

Academic Editor

PLOS ONE

Journal Requirements:

[We would like to thank the BUSCA team, especially Castrense Savojardo and Pier Luigi Martelli, for their help with the prediction of subcellular localization. This works was supported by the Liaison Committee between the Central Norway Regional Health Authority (RHA) and the Norwegian University of Science and Technology (NTNU) to [MBR]; PhD position from Enabling Technologies, Norwegian University of Science and Technology (NTNU) to [KR], the European Research Council (ERC) under the European Union‘s Horizon 2020 research and innovation program (grant agreement No 758306) [MBT] and The Norwegian Cancer society [MBT].]

 [This works was supported by the Liaison Committee between the Central Norway Regional Health Authority (RHA) and the Norwegian University of Science and Technology (NTNU) to [MBR]; PhD position from Enabling Technologies, Norwegian University of Science and Technology (NTNU) to [KR], the  European  Research  Council  (ERC)  under the  European  Union‘s Horizon 2020 research and innovation program (grant agreement No 758306) [MBT] and  The Norwegian Cancer society [MBT]. The funders had no role in study design, data collection and analysis, decision to publish, or preparation of the manuscript.]

Reviewers' comments:

Reviewer's Responses to Questions

**Comments to the Author**

1. Is the manuscript technically sound, and do the data support the conclusions?

Reviewer #1: Yes

Reviewer #2: Partly

2. Has the statistical analysis been performed appropriately and rigorously? 

Reviewer #1: Yes

Reviewer #2: I Don't Know

3. Have the authors made all data underlying the findings in their manuscript fully available?

Reviewer #1: Yes

Reviewer #2: No

4. Is the manuscript presented in an intelligible fashion and written in standard English?

Reviewer #1: Yes

Reviewer #2: Yes

5. Review Comments to the Author

Reviewer #1: 1. Is the manuscript technically sound, and do the data support the conclusions?

Manuscript is technically sound and easy to follow. Protocol could be followed and reproduced using publicly available data and tools. Use of developed methods support conclusions made about prostate cancer dataset and Kegg pathways analyzed.

2. Has the statistical analysis been performed appropriately and rigorously?

Appropriate statistical analysis was performed and provided in Table S2 regarding generation of consensus localization. Appropriate differential expression cutoffs defined in methods and applied to downloaded RNA-seq dataset.

3. Have the authors made all data underlying the findings in their manuscript fully available?

Include the original versions of the TCA cycle and Glycolysis/gluconeogenesis pathways associated with figures 2 and 3. They could be included in the supplemental to not detract from the presented mitochondrial/non-mitochondrial pathway. The original version of the Glycolysis/gluconeogenesis is referenced in both the results and discussion. Originals are already provided for Pyruvate metabolism and Alanine, glutamine, and glutamate metabolism pathways.

4. Is the manuscript presented in an intelligible fashion and written in standard English?

Fig 5b is of lower quality than a and c. It is very difficult to read.

Concerns about dual publication, research ethics, or publication ethics?

None

Reviewer #2: This is an interesting manuscript whose goal is to improve the interpretation of gene networks/pathways based on mitochondrial and non-mitochondrial localization. To do so, the authors extend their previously published algorithm called FunHop (functional homologous protein) and add information related to mitochondrial localization. To obtain this information, the authors combined experimental results with in silico results predicted using BUSCA (Bologna Unified Subcellular Component Annotator). Independently, the authors identified changes in gene expression by mining the available prostate RNA-Seq dataset. By combining these results, the authors provide four examples of the KEGG pathways. The authors show that mitochondrial proteins in the TCA cycle are upregulated in prostate cancer cells, while no significant upregulation of non-mitochondrial proteins is found in the non-mitochondrial glycolysis/gluconeogenesis pathway. Most current analyses in gene pathways and networks do not take into account subcellular localization. Therefore, this study is potentially important. I have the following suggestions for authors to further improve the manuscript.

1. A notable weakness of this study is that the results were not quantified. Although most mitochondrial genes are upregulated in the TCA cycle, other pathways may show similar numbers of upregulated and downregulated genes that belong to both mitochondrial and non-mitochondrial proteins. Can the authors develop a quantitative score to measure the entire pathway (e.g., by combining fold changes of up- or down-regulated genes)?

2. It is important to provide more details in the gene expression analysis. There are two normal cell lines and two prostate cancer (PCa) cell lines. This could yield 4 different datasets. It is not clear how the authors combined them to produce only one set of results for downstream analysis. In addition, the authors should include the results in the supplementary material.

3. A better discussion is needed when interpreting the mitochondrial and non-mitochondrial versions of the pathway. Do they imply that these reactions take place exclusively inside or outside the mitochondria? Or does the split pathway imply that these proteins need to be transferred into (or out of) mitochondria. In this case, the non-mitochondrial version of the TCA cycle would imply that these proteins need to be transferred into the mitochondria, even though they are considered non-mitochondrial proteins.

4. Related to point 3, if the idea of "transfer" is correct, then caution may be needed when integrating changes in gene expression. Even if the level of gene expression is not increased, their activity may still be increased if the level of the transfer protein is increased. Can the authors provide evidence to support this idea by examining the datasets?

5. There is little discussion of comparing the results obtained in this study with similar studies in the literature.

6. PLOS authors have the option to publish the peer review history of their article (what does this mean?). If published, this will include your full peer review and any attached files.

Reviewer #1: No

Reviewer #2: No

---

## [Author Response · Author response to Decision Letter 0]

29 Aug 2022

Response to Reviewers.

General:

First we would like to inform you that the previous Figure references 4a, 4b and 4c have been replaced by Figure references 4, 5 and 6 respectively, and that Figure references 5a, 5b and 5c have been replaced by Figure references 7, 8 and 9 respectively. The reason for this change was to present this as standalone figures rather than figure panels, since using figure panels would compromise the resolution and readability of all these figures considerably. 

Reviewer #1:

3. Have the authors made all data underlying the findings in their manuscript fully available?

Include the original versions of the TCA cycle and Glycolysis/gluconeogenesis pathways associated with figures 2 and 3. They could be included in the supplemental to not detract from the presented mitochondrial/non-mitochondrial pathway. The original version of the Glycolysis/gluconeogenesis is referenced in both the results and discussion. Originals are already provided for Pyruvate metabolism and Alanine, glutamine, and glutamate metabolism pathways.

Reply: We agree with the reviewer, and we have added the original versions of TCA cycle and Glycolysis to the Supporting Information (S4 Fig and S5 Fig respectively).

4. Is the manuscript presented in an intelligible fashion and written in standard English?

Fig 5b is of lower quality than a and c. It is very difficult to read.

Reply: We have replaced Fig 5b with a new version of better quality. It is now presented as Fig 8 (see comment above)

Reviewer #2: 

1. A notable weakness of this study is that the results were not quantified. Although most mitochondrial genes are upregulated in the TCA cycle, other pathways may show similar numbers of upregulated and downregulated genes that belong to both mitochondrial and non-mitochondrial proteins. Can the authors develop a quantitative score to measure the entire pathway (e.g., by combining fold changes of up- or down-regulated genes)?

Reply: The Supporting Information (now included with the submission) includes several plots that illustrate the distribution of mitochondrial proteins according to key properties (fraction of genes in each pathway, fraction with significant regulation, fraction with significant upregulation). We believe that these data give some of the relevant information that the reviewer asks for. 

However, “to develop a quantitative score to measure the entire pathway” that aggregates fold changes, as suggested by the reviewer, is much more complicated and outside the scope of the current project.

2. It is important to provide more details in the gene expression analysis. There are two normal cell lines and two prostate cancer (PCa) cell lines. This could yield 4 different datasets. It is not clear how the authors combined them to produce only one set of results for downstream analysis. In addition, the authors should include the results in the supplementary material.

Reply: Initially wanted to look for differences in all four cell-lines. However, when using various methods to compare cells based on differential expression results, we found that the two cancer and two normal cell-lines shared the most similarities. In addition, comparing all four cell-lines against each other would have made the analysis too comprehensive. We thus decided to focus on compartmental difference in cancer vs normal metabolism by combining the cancer and normal cell-lines. The results from differential analysis is provided in Supplemental S1 File. 

3. A better discussion is needed when interpreting the mitochondrial and non-mitochondrial versions of the pathway. Do they imply that these reactions take place exclusively inside or outside the mitochondria? Or does the split pathway imply that these proteins need to be transferred into (or out of) mitochondria. In this case, the non-mitochondrial version of the TCA cycle would imply that these proteins need to be transferred into the mitochondria, even though they are considered non-mitochondrial proteins.

Reply: The reviewer addresses a very relevant issue with this comment. In this study we have assumed that the metabolic enzymes have their activity in either mitochondrial or cytosolic compartments. Though import/export of these enzymes from the mitochondria takes place, we have not implemented this in our analysis. For example for mitochondria, most of the relevant proteins are made in the cytosol and must afterwards transported into the mitochondria, and changes in protein transport may affect the distribution of these proteins between cytosol and mitochondria. Moreover, we do not have data on whether this transport affects protein activity in the compartments. This is a limitation of our study, and we have added a brief comment about this to the discussion.

Metabolites themselves can move between compartments with the help of transport proteins. The role of transport proteins is not well modelled in current metabolic pathway schemes (like KEGG) but could be a natural and important expansion to such pathways. This includes both transport in and out of the cell itself, as well as transport between different compartments within the cell. It is clear that such transport can greatly affect the balance and availability of metabolites in the different compartments, and lack of modelling of metabolite transport is a clear limitation in metabolic pathway analysis. 

We have added a short section on compartments and transport of metabolites between compartments to the introduction, and also elaborated on issue in the discussion.

4. Related to point 3, if the idea of "transfer" is correct, then caution may be needed when integrating changes in gene expression. Even if the level of gene expression is not increased, their activity may still be increased if the level of the transfer protein is increased. Can the authors provide evidence to support this idea by examining the datasets?

Reply: As discussed above, we assume that transport that is important to our system is limited to metabolites and does not include proteins. Changes in protein transport may certainly change the picture, as indicated by the reviewer. However, we do not have access to any data that makes it possible to take this into account. We have added a short comment about this to the Discussion.

5. There is little discussion of comparing the results obtained in this study with similar studies in the literature.

Reply: We have tried to include more comparisons of our results to other studies, in particular with respect to the general increased activity of mitochondrial processes in cancers. We have added a section on this to the Discussion.

---

## [Decision Letter · Decision Letter 1]

20 Sep 2022

FunHoP analysis reveals upregulation of mitochondrial genes in prostate cancer

PONE-D-22-09961R1

Dear Dr. Rye

We’re pleased to inform you that your manuscript has been judged scientifically suitable for publication and will be formally accepted for publication once it meets all outstanding technical requirements.

Kind regards,

Thomas Burris, Ph.D.

Academic Editor

PLOS ONE

Additional Editor Comments (optional):

Reviewers' comments:

Reviewer's Responses to Questions

**Comments to the Author**

1. If the authors have adequately addressed your comments raised in a previous round of review and you feel that this manuscript is now acceptable for publication, you may indicate that here to bypass the “Comments to the Author” section, enter your conflict of interest statement in the “Confidential to Editor” section, and submit your "Accept" recommendation.

Reviewer #1: All comments have been addressed

Reviewer #2: All comments have been addressed

2. Is the manuscript technically sound, and do the data support the conclusions?

Reviewer #1: Yes

Reviewer #2: Yes

3. Has the statistical analysis been performed appropriately and rigorously? 

Reviewer #1: Yes

Reviewer #2: Yes

4. Have the authors made all data underlying the findings in their manuscript fully available?

Reviewer #1: Yes

Reviewer #2: Yes

5. Is the manuscript presented in an intelligible fashion and written in standard English?

Reviewer #1: Yes

Reviewer #2: Yes

6. Review Comments to the Author

Reviewer #1: (No Response)

Reviewer #2: The revised manuscript has addressed/clarified my previous questions and significantly improved.

I have no further questions.

7. PLOS authors have the option to publish the peer review history of their article (what does this mean?). If published, this will include your full peer review and any attached files.

Reviewer #1: No

Reviewer #2: No

---

## [Editor Report · Acceptance letter]

30 Sep 2022

PONE-D-22-09961R1 

FunHoP analysis reveals upregulation of mitochondrial genes in prostate cancer 

Dear Dr. Rye:

I'm pleased to inform you that your manuscript has been deemed suitable for publication in PLOS ONE. Congratulations! Your manuscript is now with our production department. 

Kind regards, 

on behalf of

Professor Thomas Patrick Burris 

Academic Editor

PLOS ONE